# Grape Seed Extract Eliminates Visceral Allodynia and Colonic Hyperpermeability Induced by Repeated Water Avoidance Stress in Rats

**DOI:** 10.3390/nu11112646

**Published:** 2019-11-04

**Authors:** Hideyuki Arie, Tsukasa Nozu, Saori Miyagishi, Masayuki Ida, Takayuki Izumo, Hiroshi Shibata

**Affiliations:** 1Institute for Health Care Science, Suntory Wellness Limited, Seikadai 8-1-1, Seika-cho, Soraku-gun, Kyoto 619-0284, Japan; Masayuki_Ida@suntory.co.jp (M.I.); Takayuki_Izumo@suntory.co.jp (T.I.); Hiroshi_Shibata@suntory.co.jp (H.S.); 2Department of Regional Medicine and Education, Asahikawa Medical University, Midorigaoka Higashi 2-1-1-1, Asahikawa 078-8510, Japan; tnozu@sea.plala.or.jp; 3Division of Gastroenterology and Hematology/Oncology, Department of Medicine, Asahikawa Medical University, Midorigaoka Higashi 2-1-1-1, Asahikawa 078-8510, Japan; miyagishi@asahikawa-med.ac.jp

**Keywords:** grape seed extract, visceral hypersensitivity, gut permeability, gut barrier, tight junction, inflammation, toll-like receptor 4

## Abstract

Grape seed extract (GSE) is rich in polyphenols composed mainly of proanthocyanidins, which are known to attenuate proinflammatory cytokine production. Repeated water avoidance stress (WAS) induces visceral allodynia and colonic hyperpermeability via toll-like receptor 4 (TLR4) and proinflammatory cytokine pathways, which is a rat irritable bowel syndrome (IBS) model. Thus, we explored the effects of GSE on repeated WAS (1 h for 3 days)-induced visceral allodynia and colonic hyperpermeability in Sprague-Dawley rats. Paracellular permeability, as evaluated by transepithelial electrical resistance and flux of carboxyfluorescein, was analyzed in Caco-2 cell monolayers treated with interleukin-6 (IL-6) and IL-1β. WAS caused visceral allodynia and colonic hyperpermeability, and intragastric administration of GSE (100 mg/kg, once daily for 11 days) inhibited these changes. Furthermore, GSE also suppressed the elevated colonic levels of IL-6, TLR4, and claudin-2 caused by WAS. Paracellular permeability was increased in Caco-2 cell monolayers in the presence of IL-6 and IL-1β, which was inhibited by GSE. Additionally, GSE suppressed the claudin-2 expression elevated by cytokine stimulation. The effects of GSE on visceral changes appear to be evoked by suppressing colonic TLR4-cytokine signaling and maintaining tight junction integrity. GSE may be useful for treating IBS.

## 1. Introduction

Irritable bowel syndrome (IBS) is a functional gastrointestinal disorder characterized by chronic abdominal pain with altered bowel habits [1]. Visceral hypersensitivity is considered to be one of the most important underlying mechanisms and is a hallmark of IBS [2]. Stress alters the visceral sensory function and has a significant impact on the development and exacerbation of IBS symptoms [3].

At the same time, there have been several reports showing that compromised gut barrier function manifested by gut hyperpermeability is observed in some patients with IBS, and stress also enhances gut permeability [3,4]. Impaired gut barrier leads to bacterial translocation causing increased lipopolysaccharide (LPS) and proinflammatory cytokines, which is also considered to be a pivotal feature of IBS [5,6,7]. Actually, it was reported that circulating pro-inflammatory cytokines and LPS are increased in IBS patients [5,8,9,10]. Moreover, we previously showed that LPS induced visceral hypersensitivity and gut barrier disruption via interleukin-6 (IL-6), IL-1 [7], and toll-like receptor 4 (TLR4) signaling [4].

Water avoidance stress (WAS) is a psychological stress protocol causing visceral hypersensitivity and gut hyperpermeability, and now WAS-loaded rats are generally accepted as the animal model of IBS [11]. We have recently demonstrated that IL-6, IL-1β, and TLR4 pathways mediated these visceral changes induced by repeated WAS [4,12], which is similar to the LPS-induced IBS model. These results suggest that TLR4-cytokine signaling is a significant contributor to IBS [13].

Grape seed extract (GSE) is produced as a by-product of wine and grape juice. It is rich in polyphenolic compounds including proanthocyanidins, which have anti-oxidative and anti-inflammatory properties [14,15]. These beneficial effects are considered to contribute to the prevention of cancer, cardiovascular diseases and diabetes [14,15]. Recent studies showed that GSE supplementation inhibited intestinal hyperpermeability and the expression of proinflammatory cytokines in IL-10-deficient colitis [16,17]. Furthermore, GSE was found to reduce plasma levels of LPS and proinflammatory cytokines with concomitant protection of gut barrier function mediated by tight junction (TJ) structure in animal models of obesity [18,19]. Thus, we hypothesized that GSE improves visceral sensation and gut barrier function via suppressing the TLR4-cytokine signaling and maintaining TJ structure, which are beneficial factors in the treatment of IBS.

In the present study, we investigated the effects of GSE on visceral allodynia and colonic hyperpermeability induced by repeated WAS. Moreover, the actions of GSE in epithelial monolayers *in vitro* were also evaluated.

## 2. Materials and Methods

### 2.1. Animals

This study involved seven-week-old male Sprague-Dawley rats (Charles River Laboratories, Atsugi, Japan) that weighed about 300 g. The animals were group-housed (3–4 rats/cage) in a controlled environment with a temperature of 23–25 °C and a 12/12-h light/dark cycle in the animal care facility of Asahikawa Medical University. Rats were allowed ad libitum access to food (Solid rat chow, Oriental Yeast Co., Ltd., Tokyo, Japan) and water. All studies were approved by the Research and Development and Animal Care Committees at Asahikawa Medical University (#17149, approved on 2 August 2017) and the Ethics Committee of Animal Experiment of Suntory (APRV000551, approved on 15 October 2017) in accordance with the Internal Regulations on Animal Experiments at Asahikawa Medical University and Suntory Holdings Limited, which are based on the Law for the Humane Treatment and Management of Animals (Law No. 105, 1 October 1973).

### 2.2. Grape Seed Extract

GSE was prepared from grape seeds (*Vitis vinifera Linne*) using ethanol and water as eluents and was primarily composed of 89.3% proanthocyanidins and 6.6% monomeric flavanols by dry weight (Kikkoman Biochemifa Company, Tokyo, Japan) [20]. The dose of GSE was determined according to a previous report [17].

### 2.3. Measurement of Visceral Sensation

Colonic balloon distention was performed in conscious rats to elicit abdominal muscle contractions (visceromotor response, VMR) that were measured by an electromyogram (EMG), this is a previously validated, quantitative measure of visceral nociception [7,12]. In the present study, the VMR threshold, defined as the volume (mL) of the distended balloon that induced visceral pain, was measured.

#### 2.3.1. Electrode Implantation and Colonic Distention Balloon Placement

In non-fasted rats, a small skin incision was made under anesthesia for the insertion of an electrode (Teflon-coated stainless steel, 0.05-mm diameter, MT Giken, Tokyo, Japan) approximately 2 mm into the left external oblique muscle. Cyanoacrylate instant adhesive was used to fix the electrodes to the muscle and the incised skin. The electrode leads were then directly externalized through the closed incisions. A distension balloon (6-Fr disposable silicon balloon urethral catheter, JU-SB0601, Terumo Corporation, Tokyo, Japan) was placed intra-anally such that the distal end was 2 cm proximal to the anus.

#### 2.3.2. Colonic Distention and Measurement of Abdominal Muscle Contraction

After the electrode was implanted and the balloon was placed, the rats were placed in Bollmann cages and acclimatized to the experimental conditions for a period of 30 min prior to testing. The electrode leads were then connected to an EMG amplifier. The EMG signals were amplified, filtered (3000 Hz), digitized by a PowerLab system (AD Instruments, Colorado Springs, CO, USA), and finally recorded using LabChart 7 (AD Instruments). Thirty min after the surgery, colonic distension was performed, as previously reported [12], using the ascending method of limits paradigm with phasic distensions through manual inflation of the balloon with water by syringe. The distention was progressively increased in increments of 0.1 mL every 5 s until a VMR consisting of significant sustained abdominal muscle contractions was detected (Figure 1a). The VMR threshold was assessed twice with an interval of 2 min, and the mean was calculated for each individual animal. The percentage change in threshold was calculated as the threshold value after drug administration divided by the basal threshold value and then multiplied by 100.

### 2.4. Measurement of Colonic Permeability

Colonic permeability was measured as reported previously [4]. Under anesthesia, the rats underwent laparotomy to ligate the colon at the junction with the cecum. An open-tipped catheter (3-Fr, 1-mm internal diameter, Atom, Tokyo, Japan) was then inserted into the proximal colon through the small hole made by a puncture using a needle at the 1 cm distal from the ligation and secured by a ligature. Phosphate-buffered saline (PBS) was used to gently flush the colon through the catheter to wash out all stool. The colon was additionally ligated at approximately 4 cm from the junction with the cecum, and 1 mL of 1.5% Evans blue (Sigma-Aldrich, St. Louis, MO, USA) in PBS was instilled into the colon segment through the catheter. The rats were euthanized after 15 min, their colons were excised, washed with PBS and 1 mL of 6 mM N-acetyl cysteine, opened, and then placed in 2 mL of *N,N*-dimethylformamide for 12 h. The Evans blue concentration in the supernatant was measured using a spectrophotometer at 610 nm to determine permeability.

### 2.5. Stress Protocol

The rats were exposed to WAS as reported previously [21]. Briefly, the rats were placed individually on a plastic platform (8 cm high, 6 cm long, 6 cm wide) in the middle of a plastic cage that was filled with water to 7 cm of the platform height. As controls, animals were placed in the same plastic cage, but without water (sham stress).

### 2.6. Experimental Procedures

Four groups of six to seven rats were prepared. Before subjecting the animals to WAS or sham stress, the rats underwent intragastric administration of either GSE (100 mg/kg, once daily) or vehicle for 7 days. On the 8th day, GSE or vehicle was administered again followed by measurement of the basal VMR threshold 1.5 h after administration. WAS or sham stress was then applied for 1 h. Daily stress sessions were implemented for three consecutive days, and GSE or vehicle was also administered before the stress. On the 11th day, GSE or vehicle was administered and the threshold was recorded followed by the measurement of colonic permeability 24 h after undergoing the last stress session (Figure 1b). We previously demonstrated that this repeated WAS protocol successfully induced visceral allodynia and increased colonic permeability in rats [4]. All *in vivo* experiments were done using the same animals.

### 2.7. Quantification of Colonic Cytokines

Middle colon tissues (1 cm in length) were washed and homogenized in PBS supplemented with Halt Protease and Phosphatase Inhibitor Cocktail (Thermo Fisher Scientific, Walthman, MA, USA). Protein concentrations were measured using the Bicinchoninic acid method (Thermo Fisher Scientific). Cytokine concentrations were analyzed using a V-PLEX Proinflammatory Panel 2 Rat Kit (Meso Scale Discovery, Gaithersburg, MD, USA), which is a highly sensitive multiplex enzyme-linked immunosorbent assay.

### 2.8. Quantification of Colonic Protein Expression

Colon tissues were homogenized in lysis buffer with 1% (w/v) sodium dodecyl sulfate, 1% (w/v) sodium deoxycholate and 1% (v/v) Triton X-100 in 30 mM tris (hydroxymethyl) aminomethane containing protease and phosphatase inhibitors (pH 7.4). Caco-2 cell monolayers were lysed with radio-immunoprecipitation assay buffer including protease and phosphatase inhibitors. Extracted proteins were analyzed using the Wes capillary western blot system (ProteinSimple). Primary antibodies used wereclaudin-2, claudin-3, claudin-7 and zonula occludens-1 (ZO-1) (Thermo Fisher Scientific), and TLR4 (abcam). Voltage-dependent anion-selective channel 1 (abcam) was used as a loading control.

### 2.9. Experimental Procedures in Caco-2 Cells

Caco-2 cells acquired from the European Collection of Authenticated Cell Cultures were proliferated under basic cell culture conditions [22]. Cells were seeded on transwell inserts (0.4 mM pore size, polyester, Corning, Corning, NY, USA), placed in 12-well plates and maintained for 21 days. Then cells were treated with 10 ng/mL recombinant human IL-6 and IL-1β (PeproTech, Rochy Hill, NJ, USA) in the presence or absence of 3 ug/mL GSE. Transepithelial electrical resistance (TEER) and unidirectional flux of carboxyfluorescein (Sigma-Aldrich) were analyzed in Caco-2 cell monolayers 72 h after the treatments. TEER was measured as previously described by a Millicell ERS Voltohmmeter (Merck Millipore) [22]. Carboxyfluorescein (final concentration, 20 μM) was added into the apical well 48 h after treatment with GSE and fluorescence in the basal well was analyzed 24 h later using a fluorometer (excitation 492 nm, emission 517 nm). Six Caco-2 cell monolayers were analyzed in each group.

### 2.10. Statistical Analysis

Statistical analyses were conducted with the use of IBM SPSS statistics 25 (IBM, Armonk, NY, USA). All data were presented as mean ± S.E.M. Multiple comparisons were performed using Dunnett’s test. Correlation analysis was assessed by Pearson’s test. *p* < 0.05 were considered to indicate statistical significance.

## 3. Results

### 3.1. Grape Seed Extract Prevents WAS-Induced Visceral Allodynia and Colonic Hyperpermeability

WAS significantly decreased the threshold of VMR, and GSE suppressed this response (Figure 2a). Consistent with a previous report [4], WAS increased colonic permeability, which was inhibited by GSE (Figure 2b).

### 3.2. Grape Seed Extract Attenuates the Increases in Colonic Proinflammatory Cytokine and TLR4 and Restores Tight Junction Protein Expression

Colonic levels of IL-6 and IL-1β were significantly increased by WAS (Figure 3a,b), and GSE significantly suppressed the increase in IL-6. Additionally, GSE also tended to inhibit the elevation of IL-1β levels, but not significantly. Moreover, GSE treatment attenuated the WAS-induced increase in colonic TLR4 expression (Figure 3c,d).

TJ structure plays an important role in the intestinal barrier function. Claudin-2 is known to induce leaky gut barrier [22], and claudin-3, claudin-7, ZO-1 are markers of a fully functional intestinal barrier [23,24,25]. WAS increased claudin-2 and concomitantly decreased claudin-3 expression (Figure 4a–c). GSE administration significantly inhibited the WAS-induced increase in claudin-2 expression. There were no significant changes in the levels of claudin-7 and ZO-1 following WAS (Figure 4a,d,e).

Next, we investigated the relationships between colonic permeability and the parameters measured in rats used in all experiments. Consistent with a previous report [26], colonic permeability was negatively correlated with the VMR threshold (Figure 5a). Interestingly, colonic permeability was positively correlated with the levels of IL-6, IL-1β, TLR4, and claudin-2 (Figure 5b–e).

### 3.3. Grape Seed Extract Alleviates Cytokine-Induced Increases in Epithelial Permeability in Caco-2 Cell Monolayers

Since WAS elevated colonic levels of IL-6 and IL-1β, the effects of these cytokines on epithelial permeability in Caco-2 cell monolayers were investigated. Cytokine stimulation decreased TEER and increased the unidirectional flux of carboxyfluorescein, indicating the disruption of TJ integrity (Figure 6a,b). In addition, claudin-2 expression was significantly elevated, while the expression of claudin-3 and claudin-7 was reduced (Figure 6c–f). GSE blocked the observed cytokine-induced changes (Figure 6c–f). In contrast, ZO-1 expression was not significantly altered (Figure 6g).

## 4. Discussion

We previously demonstrated that repeated WAS induced visceral allodynia and colonic hyperpermeability, which were mediated via TLR4 and proinflammatory cytokine signaling [4,12]. The current study showed that repeated WAS increased the colonic levels of IL-6, IL-1β and TLR4, and altered colonic TJ protein expression, which could impair gut barrier integrity. These results suggest that repeated WAS activates the TLR4-proinflammatory cytokine signaling to cause these visceral changes, which further supports our previous findings above.

The TLR4 pro-inflammatory cytokine system is thought to be involved in the pathophysiology of a certain proportion of IBS patients [5,8,9,10]. TLR4 in the colonic tissue of IBS patients is elevated [27], and TLR4 messenger RNA expression in the colonic mucosa is significantly correlated with the duration of symptoms in IBS patients [28]. Although it has not yet been precisely determined how the elevated expression of TLR4 in the colon contributes to the pathophysiology, cytokine production triggered by TLR4 activation might evoke the visceral functional changes observed in IBS. We have also recently demonstrated that peripheral injection of IL-6 or IL-1β induced visceral allodynia [7], possibly through the activation of cytokine receptors located in visceral afferent neurons [29,30]. Additionally, proinflammatory cytokines also increase gut permeability via regulating TJ protein expression [22]. In the present study, we showed that the colonic level of IL-6 or IL-1β was positively correlated with elevated colonic permeability, which is consistent with the evidence above.

TJ proteins are the main regulator of paracellular permeability. Knockdown of occludin which was reported to maintain TJ structure causes intestinal hyperpermeability with visceral hypersensitivity in mice, suggesting that an intestinal barrier mediated by TJ proteins plays a significant role in the development of visceral pain [31]. In the present study, we showed that repeated WAS significantly increased claudin-2, which promotes leaky gut barriers [22] and decreased claudin-3, which is involved in the maintenance of barrier function [23]. These changes appear to be associated with increased colonic permeability in the present study. On the other hand, other tested TJ proteins, such as claudin-7 and ZO-1, were unchanged. Since only claudin-2 was significantly correlated with colonic permeability among these TJ proteins, claudin-2 is considered to be the most significant contributor in our stress model. Notably, it was shown that claudin-2 expression is higher in the ileum of IBS patients than healthy controls [32].

Gut hyperpermeability leads to bacterial translocation and results in immune system activation and subsequent inflammation [33,34]. In this process, LPS is released and proinflammatory cytokines are produced through LPS activation of TLR4 [5], thereby further stimulating the TLR4-cytokine system and leading to a vicious cycle [4]. In this scenario, visceral hypersensitivity appears to result from colonic inflammation induced by gut barrier impairment. We showed that the reduced threshold of VMR was significantly related to increased colonic permeability, which supports the validity of our hypothesis.

Incidentally, the corticotropin-releasing factor (CRF) modulates the gastrointestinal stress response and may significantly contribute to the pathophysiology of IBS [3,35]. We previously demonstrated that repeated WAS-induced visceral changes were mediated via peripheral CRF receptors and TLR4, and peripheral administration of CRF mimicked these responses [4,12]. Moreover, CRF increased TLR4 expression in macrophages [36] and enhanced inflammatory cytokine production following LPS treatment [37], which possibly stimulates visceral afferent nerves. In addition, other researchers showed that CRF increased claudin-2 by upregulation of TLR4 in intestinal epithelial cells *in vitro* [38]. These findings suggest that activating peripheral CRF signaling triggered by stress modulates TLR4 signaling, which may explain the mechanism of visceral changes induced by repeated WAS.

It was demonstrated that GSE inhibited inflammatory responses via nuclear factor κB (NFκB) signaling in LPS-stimulated macrophages [39,40]. Moreover, GSE supplementation improves colonic tissue damage, which is probably mediated by inhibiting inflammatory cytokine gene expression and NFκB signaling in IL-10 KO colitis mice [16]. Therefore, we hypothesized that GSE attenuates the visceral changes triggered by repeated WAS through suppression of cytokine signaling, which was demonstrated to occur. GSE inhibited the increases in colonic IL-6 and TLR4 levels following stress. This is the first report showing the beneficial effects of GSE in a rat IBS model. Since NFκB is a key regulator of cytokine production and TLR4 expression in immune cells, GSE likely attenuates the TLR4-cytokine signaling to evoke the effects via inhibition of the NFκB pathway [41,42].

We also found that GSE suppressed the elevated expression of claudin-2 *in vivo*, which appears to be a direct mechanism of colonic barrier improvement in our IBS model, as previously described. Since IL-6 increases gut permeability by increasing expression of claudin-2 [22], one of the mechanisms underlying GSE-mediated suppression of WAS-induced elevation of claudin-2 may be attributable to the inhibition of cytokine production.

Since GSE blocked the visceral changes *in vivo*, we also explored the effects *in vitro* using Caco-2 cells stimulated by IL-6 and IL-1β. These cytokines induced hyperpermeability and increased claudin-2 expression in Caco-2 cell monolayers. Unlike the results from the *in vivo* animal model in the present study, the treatment reduced the level of claudin-7, which could also increase the permeability [25]. Since factors other than inflammatory cytokines, such as CRF or mast cells, may influence the tightness of the intestinal TJ barrier [4,43,44], the changes in TJ protein expression are thought to differ from the results *in vivo*. Interestingly, GSE attenuated the increased paracellular permeability and also fully reversed the elevated claudin-2 expression. Suzuki et al. [22] demonstrated that the IL-6-induced expression of claudin-2 in Caco-2 cells was mediated via intracellular signalings such as mitogen-activated protein kinase, kinase 1/extracellular signal-regulated kinase 1 and phosphoinositide 3-kinase. Although the cellular mechanisms of GSE remain to be elucidated, our results indicate that GSE inhibits the release of proinflammatory cytokines, as well as cytokine-induced changes in TJ protein levels, which may account for the strong effects of GSE on the visceral changes induced by repeated WAS.

The GSE used in this study contained proanthocyanidins at a level of >80%. It was reported that GSE composed of more than 85% proanthocyannidins inhibited LPS-induced NFκB signaling in macrophages [40]. Additionally, it was shown that proanthocyanidins (>95% purity) extracted from grape seeds attenuated intestinal inflammation and NFκB activation in a drug-induced colitis model [45]. Based on these lines of evidence, proanthocyanidins appear to be the main contributor to the effects of GSE in the present study, likely through the suppression of NFκB signaling.

The study has several limitations. The precise cellular or molecular mechanisms of GSE effects have not been determined. In addition, we did not show direct evidence that GSE inhibits NFκB signaling. Since the source of cytokines contributing to the visceral changes triggered by repeated WAS has not yet been determined, the cellular target of GSE also remains unknown. Further studies are needed to explore these issues.

Despite the above limitations, our results strongly suggest that GSE ameliorates the symptoms of IBS by improving visceral sensation and gut barrier integrity. Since GSE has been widely utilized as a supplement/medicine for blood vessel protection, its utilization for treating IBS is not expected to be challenging. Large-scale clinical trials that aim to evaluate the effectiveness of GSE in patients with IBS should be conducted in the future

## 5. Conclusions

GSE ameliorated repeated WAS-induced visceral allodynia and colonic hyperpermeability via inhibition of TLR4 and proinflammatory cytokine signaling with improving TJ integrity. Thus, GSE may be useful for the treatment of IBS.

## Figures and Tables

**Figure 1 nutrients-11-02646-f001:**
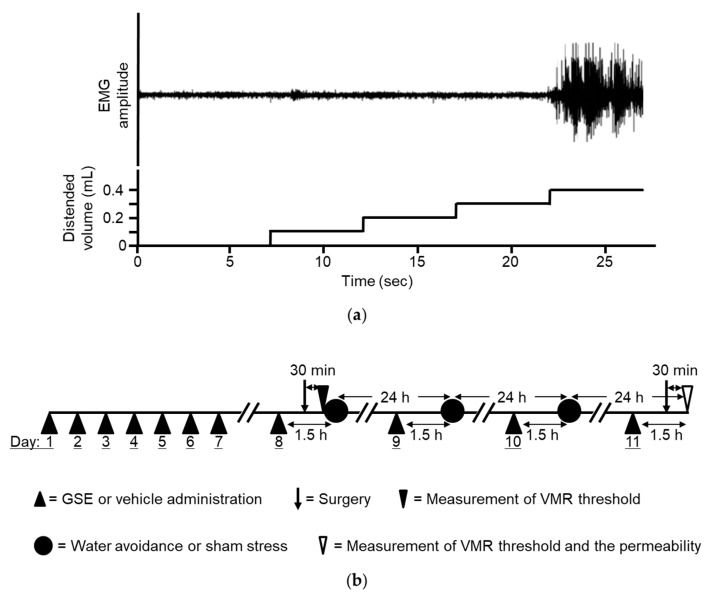
(**a**) Threshold of visceromotor response (VMR) determined by the distended balloon volume (mL) inducing apparent sustained abdominal muscle contractions. Demonstrable electromyogram (EMG) recording is depicted. The threshold of VMR was 0.4 mL in this rat. (**b**) Schematic representation of the experimental protocol. Grape seed extract (GSE) or vehicle was administered intragastrically once daily for 7 days. On the 8th day, GSE or vehicle was administered again, and 1 h later, surgical electrode implantation and insertion of a distension balloon into the distal colon were performed. The basal threshold of the VMR was analyzed 30 min after the surgery. The animals were then subjected to either WAS or sham stress for 1 h. The rats underwent this treatment, i.e., GSE or vehicle administration followed by a stress session, for 3 days. On the 11th day, the drug was administered and the analysis of the second threshold of VMR followed by the permeability test was conducted at 24 h after completing the last stress session. GSE, grape seed extract. VMR, visceromotor response, WAS, water avoidance stress.

**Figure 2 nutrients-11-02646-f002:**
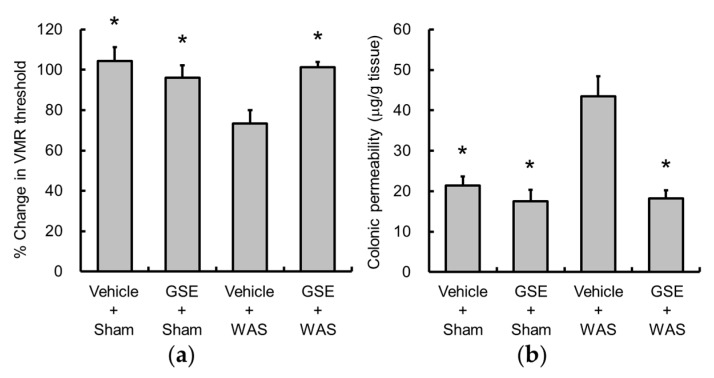
Effects of GSE on water avoidance stress (WAS)-induced visceral allodynia (**a**) and colonic hyperpermeability (**b**). GSE reversed the changes induced by WAS. Data are presented as the mean ± S.E.M (*n* = 6–7). * *p* < 0.05 vs. Vehicle + WAS group. GSE, grape seed extract, WAS, water avoidance stress.

**Figure 3 nutrients-11-02646-f003:**
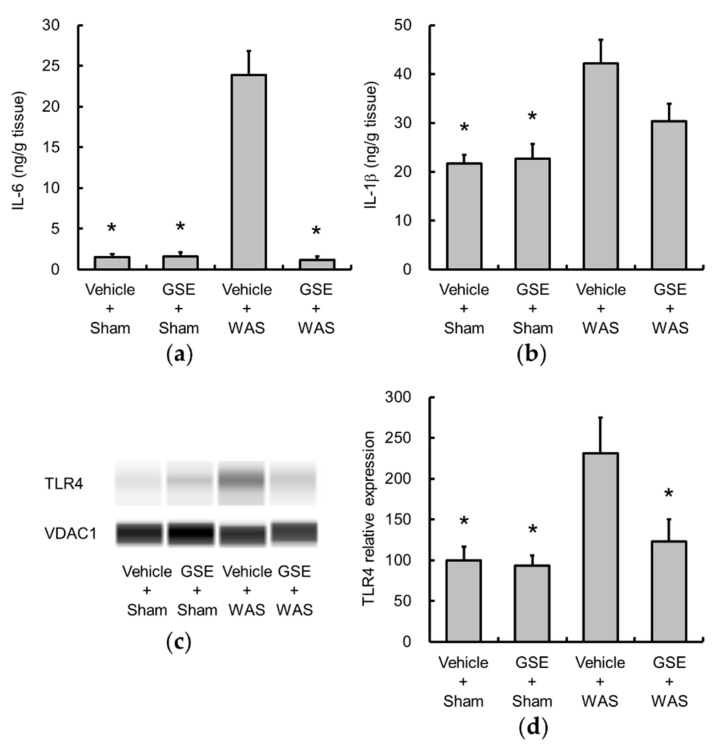
Effect of GSE on WAS-induced colonic proinflammatory cytokine and TLR4 changes. The production of IL-6 (**a**) and IL-1β (**b**) was determined by enzyme-linked immunosorbent assay, and TLR4 (**c,d**) protein expression was evaluated by capillary western blot and normalized by VDAC1. The levels of IL-6 (**a**), IL-1β (**b**) and TLR4 (**d**) were elevated by WAS, and GSE suppressed the increases in IL-6 and TLR4. Representative images are shown (**c**). Data are presented as the mean ± S.E.M (*n* = 6–7). * *p* < 0.05 vs. Vehicle + WAS group. GSE, grape seed extract, IL, interleukin, TLR4, toll-like receptor 4, VDAC1, Voltage-dependent anion-selective channel 1, WAS, water avoidance stress.

**Figure 4 nutrients-11-02646-f004:**
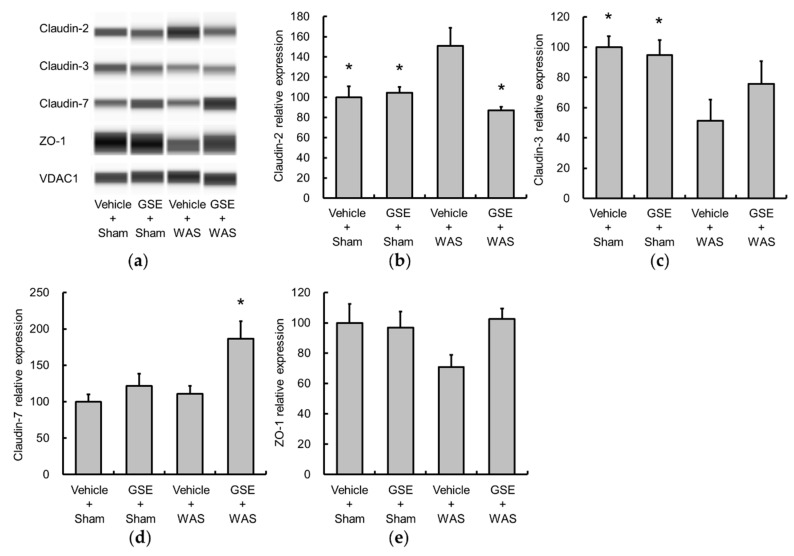
Effects of GSE on WAS-induced changes in tight junction (TJ) protein expression. Colonic TJ protein expression was evaluated using a ProteinSimple Wes system and normalized by VDAC1. Representative images are indicated (**a**). WAS increased claudin-2 (**b**) and reduced claudin-3 (**c**), and GSE prevented the elevation of claudin-2 levels. The expressions of claudin-7 (**d**) and ZO-1 (**e**) were not altered by WAS. Data are presented as the mean ± S.E.M (*n* = 6–7). * *p* < 0.05 vs. Vehicle + WAS group. GSE, grape seed extract, TJ, tight junction, VDAC1, Voltage-dependent anion-selective channel 1, WAS, water avoidance stress, ZO-1, zonula occludens-1.

**Figure 5 nutrients-11-02646-f005:**
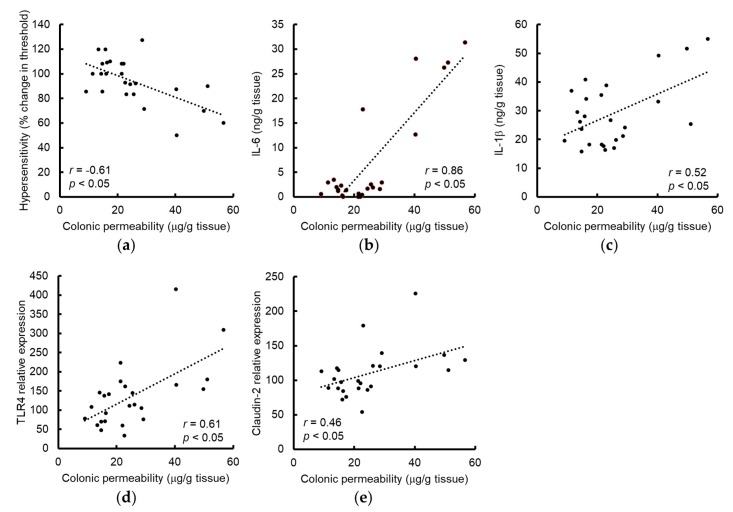
Correlations between colonic permeability and the parameters evaluated in rats from all treatment groups administered vehicle or GSE with or without WAS. The threshold of VMR showed a negative correlation with intestinal permeability (**a**). The level of IL-6 (**b**), IL-1β (**c**), TLR4 (**d**) or claudin-2 (**e**) were positively correlated to intestinal permeability. Inset indicates Pearson’s *r* correlation and corresponding *p*-value. GSE, grape seed extract, IL, interleukin, TLR4, toll-like receptor 4, VMR, visceromotor response, WAS, water avoidance stress.

**Figure 6 nutrients-11-02646-f006:**
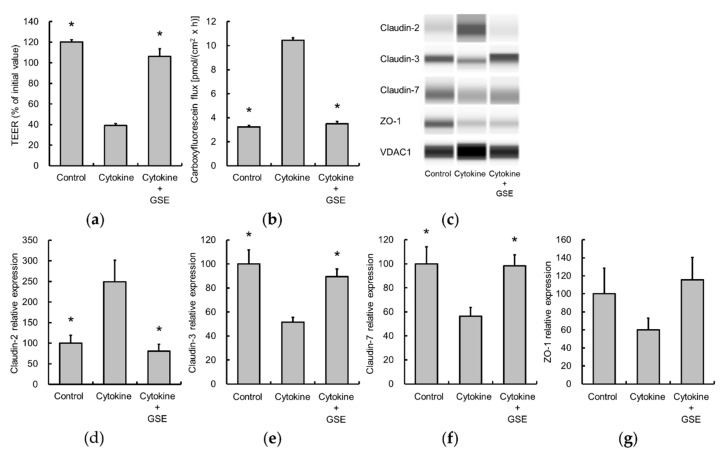
Effect of GSE on disruption of epithelial barrier function by IL-6 and IL-1β treatment in Caco-2 cell monolayers. GSE treatment reverted the barrier dysfunction as determined by TEER (**a**) and carboxyfluorescein flux (**b**) in the presence of cytokines (10 ng/mL). TJ protein expression was evaluated using a ProteinSimple Wes system and normalized VDAC1. Representative images are indicated (**c**). Cytokines increased claudin-2 (**d**) and decreased claudin-3 (**e**) and claudin-7 (**f**), which were restored by GSE. ZO-1 expression was not changed significantly (**g**). Data are presented as the mean ± S.E.M of six monolayers in each group. * *p* < 0.05 vs. Cytokine group. GSE, grape seed extract, IL, interleukin, TEER, transepithelial electrical resistance, TJ, tight junction, VDAC1, Voltage-dependent anion-selective channel 1, ZO-1, zonula occludens-1.

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
