# Peer review of "Grape Seed Extract Eliminates Visceral Allodynia and Colonic Hyperpermeability Induced by Repeated Water Avoidance Stress in Rats"

_nutrients, 2019, doi:10.3390/nu11112646_

Round 1

Reviewer 1 Report

The manuscript “Grape seed extract inhibits visceral allodynia ….” by Arie et al investigates the effects of GSE on repeated water avoidance stress -induced visceral hypersensitivity and colonic permeability in Sprague-Dawley rats. The authors infer that intragastric administration of GSE is able to inhibit WAS-induced visceral allodynia and colonic permeability. Overall the study is well designed and executed, and sufficiently detailed.

Major comment:

Only bar graphs are shown; please show the western blot images with the graphs.

Minor comment:

The legends are not detailed. Please include the method type that was executed to arrive at the data. The Reviewer had to frequently refer to the methods section to figure out what method was performed.

Author Response

We appreciated critical review comments, which greatly improved an earlier draft of this manuscript.

Major comment:

Only bar graphs are shown; please show the western blot images with the graphs.

Thank you for the comment. I added the images in Fig. 3c, Fig. 4a and Fig. 6c. And also added the explanation in Figure legends as follows,

“Representative images are shown (c).” Line197

“Representative images are indicated (a).” Line208

“Representative images are indicated (c).” Line234-235

Minor comment:

The legends are not detailed. Please include the method type that was executed to arrive at the data. The Reviewer had to frequently refer to the methods section to figure out what method was performed.

According to the comments, I added the descriptions to the figure legends regarding the method.

In Figure 3. “The production of IL-6 (a) and IL-1b (b) was determined by enzyme-linked immunosorbent assay, and TLR4 (c, d) protein expression was evaluated by capillary western blot and normalized by VDAC1. The levels of IL-6 (a), IL-1β (b) and TLR4 (d) were elevated by WAS, and GSE suppressed the increases in IL-6 and TLR4.” Line193-195

In Figure 4. “Colonic TJ protein expression was evaluated using a ProteinSimple Wes system and normalized by VDAC1.” Line206-207

In Figure 6. “TJ protein expression was evaluated using a ProteinSimple Wes system and normalized VDAC1.” Line233-234

Reviewer 2 Report

The MS by Arie et al. describes the effects of grape seed extracts (GSEs) in inhibiting visceral allodynia and increased colon permeability after water avoidance stress in rat. Based on EMG results for assessing visceromtor responses (VMR), quantification of colonic cytokines, western blot analysis and a correlative study in vitro on Caco-2 cells, authors conclude that GSEs may be useful for treating the irritable bowel syndrome (IBS) in rats.

The MS is of potential interest and could be published in Nutrients after major revision.

Main points

The chemical composition of GSE should be determined in both qualitative and quantitative terms. GSEs are very heterogeneous according to the seeds of origin and this is a crucial point to be clarified, as there is no info about this also in ref. 17.

The experimental paradigm used is very heavy and raises some ethical concerns. Why GSE has not added to food or water rather than being given intragastrically?

The results of western blots should be shown not only in statistical analysis

There is a complete lack of histological analysis that would be necessary to confirm the establishment of IBS as a consequence of WAS. Measurement of VMR could be influenced previous surgery, also considering the limited % changes in a quite limited number of mice (6-7 if well understood) and a quite high p value (0.05).

Representative images of the expression of claudins and ZO-1 in Caco2 cells should be provided. N=6 in these experiments means what? Plates? Did you check for number/concentration of cells in wells? This should be done for proper quantiation.

Minor

Number of animals used should be explicitly clarified in M&M. The 6-7 animals were used in all different types of experiments?

Line 83 Measurement of visceral sensation: As correctly stated at line 86 what is measured is VMR. This is not  sensation that occurs at cortical level! Please correct accordingly throughout MS if necessary

Author Response

We appreciated critical review comments, which greatly improved an earlier draft of this manuscript.

Main points

The chemical composition of GSE should be determined in both qualitative and quantitative terms. GSEs are very heterogeneous according to the seeds of origin and this is a crucial point to be clarified, as there is no info about this also in ref. 17.

Thank you for your variable comment. I added the description regarding the composition of GSE and its reference, as follows,

“GSE was prepared from grape seeds (Vitis vinifera Linne) using ethanol and water as eluents, and was composed of 89.3% proanthocyanidins and 6.6% monomeric flavanols by dry weight (Kikkoman Biochemifa Company) (Yamakoshi et al., 2001).” Line78-79

The experimental paradigm used is very heavy and raises some ethical concerns. Why GSE has not added to food or water rather than being given intragastrically?

Intragastrical administration by gavage is widely used method. The article using this method was also published in Nutrients (Bohmdorfer et al., 2017). Thus, there seems to be no ethical concern on this procedure, and our all studies were approved by the Animal Care Committees of our institutions, which are based on the Law for the Humane Treatment and Management of Animals, as described in the previous version of manuscript. The reason why I chose intragastrical administration by gavage is to perform the administration of precise dose of GSE. As stress possibly alters food and water consumption more or less, I presumed that GSE was not able to be administered to the animals with a preset dose precisely using the food or water containing GSE.

The results of western blots should be shown not only in statistical analysis

Thank you for the comment. I added the images in Fig. 3c, Fig. 4a and Fig. 6c. And also added the explanation in Figure legends as follows,

“Representative images are shown (c).” Line197

“Representative images are indicated (a).” Line208

“Representative images are indicated (c).” Line234-235

There is a complete lack of histological analysis that would be necessary to confirm the establishment of IBS as a consequence of WAS.

Thank you for your valuable comment. It is well known that routine histologic examination reveals no colonic mucosal abnormality in majority of the patients with IBS (Kirsch and Riddell, 2006). Thus, an animal IBS model should not display any significant histological abnormality in the colon. At the same time, previous report demonstrated that repeated WAS (10 days) induced visceral hypersensitivity but neither evoked any significant difference in the structural histology of the colon nor in the number of polymorphonuclear cells in hematoxylin and eosin-stained sections of the submucosa plus mucosa layer of colonic samples in rats (Bradesi et al., 2005). This is the reason why repeated WAS model is generally accepted to be an animal IBS model. Although we did not evaluate the histology, our model (repeated WAS for 3 days) is also considered not to develop significant histological changes. On the other hand, some reports showed that mast cell numbers in the colonic mucosa was slightly increased in IBS and repeated WAS model detected by immunohistochemistry (Bradesi et al., 2005; Kirsch and Riddell, 2006). In this context, it is not denied the possibility that mast cells count increased in the current study settings, but our study did not focus on the mast cells itself. Thus, I do not think that there is merit to strengthen the results of our study.

Measurement of VMR could be influenced previous surgery, also considering the limited % changes in a quite limited number of mice (6-7 if well understood) and a quite high p value (0.05).

Our procedures are considered not to be significant impact on visceral sensation in the current experimental settings. We published several articles (Nozu et al., 2017a, 2019a; Nozu et al., 2016, 2017b, c, 2018a, b, 2019b; Okumura et al., 2018a, b; Okumura et al., 2015a, b, 2016a, b; Okumura et al., 2018c), using the same method assessing visceral sensation. The threshold at baseline and 3 days later in sham-stressed rats was not different as shown in the previous version of manuscript. Besides we also confirmed the evidence in the preliminary experiment that the threshold in age-matched rats underwent single or two times surgery was not different either. Furthermore, the number of animals was quite limited as the reviewer pointed out, but the data regarding visceral sensation was quite clear, i.e. WAS reduced the threshold of VMR and GSE reversed the response. Moreover, sham stress or vehicle per se did not alter the threshold.

In addition, the surgery for the electrodes implantation for EMG is needed in order to assess visceral pain by measuring abdominal muscle contractions in response to colonic balloon distention in rats. It is essential procedure. Majority of the studies assessing visceral sensation used chronically implanted electrodes, which is different from our method, i.e. acute implantation of electrodes. In this method, the electrode-leads are needed to pass through subcutaneous tunnel in order to avoid damaging electrodes by animal bite. Therefore, the operational damage is considered to be greater as compared with our method (because subcutaneous tunnel is not made in our method). Additionally, the surgery must be performed several days before the experiments and need single housing until the experiments in order to prevent bites by other animals (Bradesi et al., 2006; Bradesi et al., 2005; Gilet et al., 2014; Hong et al., 2009; Hong et al., 2015; Xu et al., 2014). The single housing was reported to give crucial impact to visceral sensory function in rodents (Larauche et al., 2010). Our method is acute animal preparation without single housing to avoid this drawback. Although our method only needs minor skin incision without subcutaneous tunnel for the electrodes implantation, repeated surgery might induce some influence in immune system, as the reviewer mentioned, which could modify the results. It is not known which factor such as single housing or repeated minor surgery gives a significant impact in visceral sensory response. These results suggest our method may be rational.

Representative images of the expression of claudins and ZO-1 in Caco2 cells should be provided. N=6 in these experiments means what? Plates? Did you check for number/concentration of cells in wells? This should be done for proper quantiation.

According to the comments, I added the images in Figure 4 and the descriptions regarding the method as follows,

Six Caco-2 cell monolayers were analyzed in each group.” Line172-173.

“Data are presented as the mean ± S.E.M of six monolayers in each group.” Line236-237.

Minor

Number of animals used should be explicitly clarified in M&M. The 6-7 animals were used in all different types of experiments?

I described the number of animals used in vivo experiments in M&M as follows,

“Four groups of six to seven rats were prepared”. Line126

“All in vivo experiments were done using same animals.” Line134-135

Line 83 Measurement of visceral sensation: As correctly stated at line 86 what is measured is VMR. This is not sensation that occurs at cortical level! Please correct accordingly throughout MS if necessary

We detected VMR by EMG. Visceral sensation was assessed by measuring the threshold of VMR. The threshold of VMR was defined as the distended balloon volume (mL) inducing VMR. In order to explain the method more precisely and to avoid misunderstanding, we improved the explanation as follows,

“Colonic balloon distention was performed in conscious rats to elicit abdominal muscle contractions (visceromotor response; VMR) that were measured by an electromyogram (EMG); this is a previously validated, quantitative measure of visceral nociception [7, 12]. In the present study, the VMR threshold, defined as the volume (mL) of the distended balloon that induced visceral pain, was measured.” Line83-87.

And we also added the figure depicting the VMR and the threshold in Figure 1a, and the description in Figure legends, Figure 1,

“(a) Threshold of VMR determined by the distended balloon volume (mL) inducing apparent sustained abdominal muscle contractions. Demonstrable EMG recording is depicted. The threshold of VMR was 0.4 mL in this rat.” Line136-138.

References

Bohmdorfer, M., Szakmary, A., Schiestl, R.H., Vaquero, J., Riha, J., Brenner, S., Thalhammer, T., Szekeres, T., Jager, W., 2017. Involvement of UDP-Glucuronosyltransferases and Sulfotransferases in the Excretion and Tissue Distribution of Resveratrol in Mice. Nutrients 9, 10.3390/nu9121347.

Bradesi, S., Kokkotou, E., Simeonidis, S., Patierno, S., Ennes, H.S., Mittal, Y., McRoberts, J.A., Ohning, G., McLean, P., Marvizon, J.C., Sternini, C., Pothoulakis, C., Mayer, E.A., 2006. The role of neurokinin 1 receptors in the maintenance of visceral hyperalgesia induced by repeated stress in rats. Gastroenterology 130, 1729-1742.

Bradesi, S., Schwetz, I., Ennes, H.S., Lamy, C.M., Ohning, G., Fanselow, M., Pothoulakis, C., McRoberts, J.A., Mayer, E.A., 2005. Repeated exposure to water avoidance stress in rats: a new model for sustained visceral hyperalgesia. Am. J. Physiol. Gastrointest. Liver Physiol. 289, G42-53.

Gilet, M., Eutamene, H., Han, H., Kim, H.W., Bueno, L., 2014. Influence of a new 5-HT4 receptor partial agonist, YKP10811, on visceral hypersensitivity in rats triggered by stress and inflammation. Neurogastroenterol. Motil. 26, 1761-1770.

Hong, S., Fan, J., Kemmerer, E.S., Evans, S., Li, Y., Wiley, J.W., 2009. Reciprocal changes in vanilloid (TRPV1) and endocannabinoid (CB1) receptors contribute to visceral hyperalgesia in the water avoidance stressed rat. Gut 58, 202-210.

Hong, S., Zheng, G., Wiley, J.W., 2015. Epigenetic regulation of genes that modulate chronic stress-induced visceral pain in the peripheral nervous system. Gastroenterology 148, 148-157 e147.

Kirsch, R., Riddell, R.H., 2006. Histopathological alterations in irritable bowel syndrome. Mod. Pathol. 19, 1638-1645.

Larauche, M., Gourcerol, G., Million, M., Adelson, D.W., Taché, Y., 2010. Repeated psychological stress-induced alterations of visceral sensitivity and colonic motor functions in mice: influence of surgery and postoperative single housing on visceromotor responses. Stress 13, 343-354.

Nozu, T., Miyagishi, S., Kumei, S., Nozu, R., Takakusaki, K., Okumura, T., 2017a. Lovastatin inhibits visceral allodynia and increased colonic permeability induced by lipopolysaccharide or repeated water avoidance stress in rats. Eur. J. Pharmacol. 818, 228-234.

Nozu, T., Miyagishi, S., Kumei, S., Nozu, R., Takakusaki, K., Okumura, T., 2019a. Metformin inhibits visceral allodynia and increased gut permeability induced by stress in rats. J. Gastroenterol. Hepatol. 34, 186-193.

Nozu, T., Miyagishi, S., Nozu, R., Takakusaki, K., Okumura, T., 2016. Water avoidance stress induces visceral hyposensitivity through peripheral corticotropin releasing factor receptor type 2 and central dopamine D2 receptor in rats. Neurogastroenterol. Motil. 28, 522-531.

Nozu, T., Miyagishi, S., Nozu, R., Takakusaki, K., Okumura, T., 2017b. Lipopolysaccharide induces visceral hypersensitivity: role of interleukin-1, interleukin-6, and peripheral corticotropin-releasing factor in rats. J. Gastroenterol. 52, 72-80.

Nozu, T., Miyagishi, S., Nozu, R., Takakusaki, K., Okumura, T., 2017c. Repeated water avoidance stress induces visceral hypersensitivity; role of IL-1, IL-6 and peripheral corticotropin-releasing factor. J. Gastroenterol. Hepatol. 32, 1958-1965.

Nozu, T., Miyagishi, S., Nozu, R., Takakusaki, K., Okumura, T., 2018a. Altered colonic sensory and barrier functions by CRF: roles of TLR4 and IL-1. J. Endocrinol. 239, 241-252.

Nozu, T., Miyagishi, S., Nozu, R., Takakusaki, K., Okumura, T., 2018b. Pioglitazone improves visceral sensation and colonic permeability in a rat model of irritable bowel syndrome. J. Pharmacol. Sci. 139, 46-49.

Nozu, T., Miyagishi, S., Nozu, R., Takakusaki, K., Okumura, T., 2019b. Dehydroepiandrosterone sulfate improves visceral sensation and gut barrier in a rat model of irritable bowel syndrome. Eur. J. Pharmacol.

Okumura, T., Nozu, T., Kumei, S., Ohhira, M., 2018a. Central oxytocin signaling mediates the central orexin-induced visceral antinociception through the opioid system in conscious rats. Physiol. Behav. 198, 96-101.

Okumura, T., Nozu, T., Kumei, S., Ohhira, M., 2018b. Role of the cannabinoid signaling in the brain orexin- and ghrelin-induced visceral antinociception in conscious rats. J. Pharmacol. Sci. 137, 230-232.

Okumura, T., Nozu, T., Kumei, S., Takakusaki, K., Miyagishi, S., Ohhira, M., 2015a. Antinociceptive action against colonic distension by brain orexin in conscious rats. Brain Res. 1598, 12-17.

Okumura, T., Nozu, T., Kumei, S., Takakusaki, K., Miyagishi, S., Ohhira, M., 2015b. Involvement of the dopaminergic system in the central orexin-induced antinociceptive action against colonic distension in conscious rats. Neurosci. Lett. 605, 34-38.

Okumura, T., Nozu, T., Kumei, S., Takakusaki, K., Miyagishi, S., Ohhira, M., 2016a. Adenosine A1 receptors mediate the intracisternal injection of orexin-induced antinociceptive action against colonic distension in conscious rats. J. Neurol. Sci. 362, 106-110.

Okumura, T., Nozu, T., Kumei, S., Takakusaki, K., Miyagishi, S., Ohhira, M., 2016b. Levodopa acts centrally to induce an antinociceptive action against colonic distension through activation of D2 dopamine receptors and the orexinergic system in the brain in conscious rats. J. Pharmacol. Sci. 130, 123-127.

Okumura, T., Nozu, T., Kumei, S., Takakusaki, K., Ohhira, M., 2018c. Ghrelin acts centrally to induce an antinociceptive action during colonic distension through the orexinergic, dopaminergic and opioid systems in conscious rats. Brain Res. 1686, 48-54.

Yamakoshi J.; Tokutate S.; Kikuchi M.; Kubota Y.; Konishi H.; Mitsuoka T. Effect of proanthocyanidin-rich extract from grape seeds on human fecal flora and fecal odor. Microb. Ecol. Health. Dis. 2001, 13, 25-31.

Xu, D., Gao, J., Gillilland, M.r., Wu, X., Song, I., Kao, J.Y., Owyang, C., 2014. Rifaximin alters intestinal bacteria and prevents stress-induced gut inflammation and visceral hyperalgesia in rats. Gastroenterology 146, 484-496 e484.

Reviewer 3 Report

The manuscript entitled "Grape seed extract inhibits visceral allodynia and increased colonic permeability induced by repeated water avoidance stress in rats", authors are executed their study classically and using the high-throughput technique by evaluating live measurement which determines the actual value of the entire study.  However, to add strengthen this article ..

Author may show Western Blot analysis of, at least one of the Claudin and/or Zo proteins in GSE treated or untreated WAS animals colonic epithelial cells. Author might use histology score or immunohistochemistry to ensure GSE mediated attenuation and reversal of colonic inflammation.

Author Response

We appreciated critical review comments, which greatly improved an earlier draft of this manuscript.

Author may show Western Blot analysis of, at least one of the Claudin and/or Zo proteins in GSE treated or untreated WAS animals colonic epithelial cells.

Thank you for the comment. I added the images in Fig. 3c, Fig. 4a and Fig. 6c. And also added the explanation in Figure legends as follows,

“Representative images are shown (c).” Line197

“Representative images are indicated (a).” Line208

“Representative images are indicated (c).” Line234-235

Author might use histology score or immunohistochemistry to ensure GSE mediated attenuation and reversal of colonic inflammation.

It is well known that routine histologic examination reveals no colonic mucosal abnormality in majority of the patients with IBS (Kirsch and Riddell, 2006). Thus, an animal IBS model should not display any significant histological abnormality in the colon. At the same time, previous report demonstrated that repeated WAS (10 days) induced visceral hypersensitivity but did not evoke any significant difference in the structural histology of the colon and significant difference in the number of polymorphonuclear cells in hematoxylin and eosin-stained sections of the submucosa plus mucosa layer of colonic samples in rats either (Bradesi et al., 2005). This is the reason why repeated WAS model is generally accepted to be an animal IBS model. Although we did not evaluate the histology, our model (repeated WAS for 3 days) is also considered not to develop significant histological changes. We are very sorry that we used inadequate description regarding “colonic inflammation”, which may lead to misunderstanding that this model displays significant histological changes in the colon. In this context, we changed the expression.

“Grape seed extract attenuates the increases in colonic proinflammatory cytokine and TLR4 and restores tight junction protein expression” Line187-188

In Figure legends, Figure 3, “Effect of GSE on WAS-induced colonic proinflammatory cytokine and TLR4 changes.” Line193

References

Bradesi, S., Schwetz, I., Ennes, H.S., Lamy, C.M., Ohning, G., Fanselow, M., Pothoulakis, C., McRoberts, J.A., Mayer, E.A., 2005. Repeated exposure to water avoidance stress in rats: a new model for sustained visceral hyperalgesia. Am. J. Physiol. Gastrointest. Liver Physiol. 289, G42-53.

Kirsch, R., Riddell, R.H., 2006. Histopathological alterations in irritable bowel syndrome. Mod. Pathol. 19, 1638-1645.

Round 2

Reviewer 1 Report

The reviewer is satisfied with the modifications/ additions in the revised Manuscript.

Reviewer 2 Report

The authors have responded to all my observations.

Please not that line 93 should be IL-1β